# Role of JAK-STAT Pathway in Broiler Chicks Fed with Chestnut Tannins

**DOI:** 10.3390/ani11020337

**Published:** 2021-01-29

**Authors:** Annah Lee, Gabriela Cardoso Dal Pont, Michele Battaglia, Ryan J. Arsenault, Michael H. Kogut

**Affiliations:** 1Department of Poultry Science, Texas A&M University, College Station, TX 77843, USA; annahlee89@tamu.edu (A.L.); gabrieladp@tamu.edu (G.C.D.P.); 2Silvateam s.p.a., San Michele Mondovi CN, 12080 Piedmont, Italy; michele@silvateam.com; 3Department of Animal & Food Sciences, University of Delaware, Newark, DE 19716, USA; rja@udel.edu; 4Southern Plains Agricultural Research Center, USDA-ARS, College Station, TX 77845, USA

**Keywords:** tannins, JAK-STAT, kinome, immunity

## Abstract

**Simple Summary:**

Current bans on the use of antibiotics in livestock feed have led to increasing demand for alternatives to antibiotics (ATA). One popular alternative is chestnut tannins (ChT). While there is growing evidence of the immune benefits of using phytobiotics, such as ChT, there is currently minimal information on the effect of tannins on the immune pathway of the host. A previous study discovered a large upregulation of IL-6 in broiler chicks at day 6 when provided with 1% ChT from hatch. In regard to potential immune modulation, ChT appear to influence host immunity via an IL-6 mediated response, which could be beneficial in host defenses against pathogens at the early stages of broiler growth and development. A critical pathway identified in the regulation of the immune system is the JAK-STAT signaling pathway. The role of JAK-STAT pathway is altered by the addition of ChT in the diet. By demonstrating the changes in the kinome of the broiler model, the information in this study will provide further insights into potential ATA to improve poultry health.

**Abstract:**

The objective of this study was to identify the phosphorylation events associated with host immunity with the inclusion of chestnut tannins (ChT) in the diet. A total of 200 male day-of-hatch Cobb 500 chicks were randomly assigned to two treatment groups, totaling 50 chicks per pen per experiment (this study was repeated two times). The treatments were as follows: (1) control feed—normal starter feed (n = 50), and (2) 1% ChT inclusion feed (n = 50). The ceca were collected on each necropsy day for analysis via (1) a peptide array to provide tissue immunometabolism information from the host, and (2) quantitative PCR for mRNA expression. Of the top three immune pathways, the data identified the T-cell receptor signaling pathway, the chemokine signaling pathway, and the JAK-STAT signaling pathway. The results showed significantly altered phosphorylation of JAK and STAT peptides within the JAK-STAT pathway. These results support the mRNA expression data with the upregulated IL-6 response, due to the significant phosphorylation of IL6ST, JAK, and STAT peptides. In regard to immune modulation, ChT appear to influence host immunity via an IL-6 mediated response which could be beneficial in host defenses against pathogens at the early stages of broiler growth and development. Therefore, it is suggested that the role of the JAK-STAT pathway is altered by including ChT in the diet.

## 1. Introduction

Antibiotic growth promoters (AGPs) were originally incorporated into livestock feed to promote better health and performance. However, with the ban of antibiotics in feed [1], there is a need for alternatives that will stimulate similar outcomes of AGPs. Recently, there has been increased interest in utilizing plant-based compounds, or phytobiotics, as antibiotic growth promoter alternatives, including chestnut tannins (ChT—*Castanea sativa*) [2,3]. Phytogenic compounds are viable candidates to replace AGPs due to their bioactive properties that emulate similar properties of antibiotics without causing antimicrobial resistance [4]. Plant-based tannins can be categorized into two major groups: condensed tannins and hydrolyzable tannins [5]. Tannins can be found in many plant species, mostly in the inedible portions of the plant such as the bark or wood [6,7]. This popular alternative to antibiotics (ATA) is already widely utilized on commercial farms for its overall health benefit and improved performance, although the mechanism of how it works is still not fully understood. Host gut immunity is better protected from pathogens via increased villus height:crypt depth ratio, improved intestinal mucosa to prevent tissue damage, production of pro-inflammatory and anti-inflammatory cytokines, and enhanced expression of tight junction proteins to modulate the immune system [4,8,9,10].

Studying phosphorylation events provides information on the mechanism of post-translational modification, which offers insight in cellular and tissue phenotypes [11,12]. The species-specific kinome array utilized in this study can measure the kinase activity on the tissue level, thereby allowing the observation of post-translational modifications and providing further insight into metabolic and immune function [13]. Our previous manuscript analyzed the metabolic phosphorylation events when ChT were included in the diet [14]. Therefore, the objective of this study was to analyze the outcome of specific immune phosphorylation events when broiler chicks were fed a ChT inclusion diet posthatch for up to 10 days. We analyzed multiple timepoints of chicken-specific kinomic immune changes in avian cecal tissue of chicks fed 1% ChT. Using this technique, we were able to identify specific phosphorylation-based immune post-translational signaling changes.

## 2. Materials and Methods

### 2.1. Experimental Animals, Housing, and Treatments

All experiments conducted were in accordance with guidelines set by the United States Department of Agriculture Animal Care and Use Committee (USDA ACUC #2019001), which meets all federal requirements as defined in the Animal Welfare Act, and the Humane Care and Use of Laboratory Animals. A total of 200 male day-of-hatch Cobb 500 chicks were obtained from a local commercial hatchery and assigned to two treatment groups, totaling 50 chicks per pen per experiment. The treatments were as follows: (1) control feed—corn/soy starter feed (n = 50) and (2) 1% ChT inclusion feed (n = 50). The chicks were randomly distributed into each group with fresh pine shavings, water, and the starter diet ad libitum. The chicks were kept under 96 h light and then transitioned to 18 h light and 6 h dark cycle until the end of the study. All treatments included a corn/soybean-based crumble diet, but differed in AGP or tannins inclusion. The diets were formulated to meet or exceed broiler requirements, as shown in Appendix A. The hydrolyzable ChT additive (Silvateam s.p.a. ©, Buenos Aires, Argentina) contained 75% tannin content, supplemented with 94% dry matter, lignin and sugars. Each experimental process lasted 10 days. This study was repeated a total of two separate times.

### 2.2. Sample Collection and Processing

On each necropsy day, ten birds per group were selected by random, euthanized via cervical dislocation, and necropsied on days 2, 4, 6, 8, and 10 of each separate experiment. Both ceca were removed, flushed with phosphate buffered saline (PBS), and flash frozen in liquid nitrogen to preserve the kinase enzymatic activity. The frozen tissues were stored at −80 °C until further processing. Therefore, 100 ceca were collected on each separate experimental trial, for a total of 200 ceca collected at the end of both experiments.

### 2.3. Bacterial Detection

During each necropsy, the cecal contents (0.25 g/bird) were collected and serially diluted to 1:100, 1:1000, 1:10,000, and 1:100,000 of 1× PBS. These contents were spread onto XLT4 for Salmonella detection for confirmation purposes. These plates were incubated at 37 °C for 24 h. Furthermore, an enrichment step was also performed for detection purposes: 100 uL of the cecal contents were pre-enriched in Rappaport-Vassiliadis broth for 24 h at 37 °C. None of the birds showed the presence of *Salmonella enterica* throughout the study.

### 2.4. Chicken-Specific Kinome (Peptide) Array

For the phenotype readout, a peptide array was utilized to provide tissue immunometabolism information from the host. At three of the timepoints (days 4, 6, and 10), three whole ceca from three randomly selected birds—stored at −80 °C—were defrosted for analysis. Each distal end of the cecal sample was weighed to obtain a consistent 40 mg sample for the array. The samples were homogenized using an Omni International Bead Ruptor Elite (Kennesaw, GA, USA) in 100 uL of lysis buffer (20 mM Tris–HCl pH 7.5, 150 mM NaCl,1 mM EDTA, 1 mM ethylene glycol tetraacetic acid (EGTA), 1% Triton X-100, 2.5 mM sodium pyrophosphate, 1 mM Na3VO4, 1 mM NaF, 1 μg/mL leupeptin, 1 g/mL aprotinin, and 1 mM phenylmethylsulphonyl fluoride). All products were obtained from Sigma Aldrich (St. Louis, MO, USA), unless indicated. Following homogenization, the peptide array protocol was carried out with previously described alterations [13,15]. The resulting tissue lysates were applied onto the PepStar peptide microarrays customized by JPT Peptide Technologies GmbH (Berlin, Germany).

### 2.5. Real-Time Quantitative RT-PCR Assay

Total RNA was extracted using a Qiagen RNeasy^®^ Plus kit (Germantown, MD, USA) and evaluated with a NanoDrop^TM^ 2000 Spectrophotometer (Thermo Fisher Scientific, Waltham, MA, USA). Ceca stored in RNALater were used for RNA isolation with the Qiagen Rneasy Plus Kit. The ceca were cut longitudinally to expose the lumen, and any remaining fecal matter was gently removed with forceps. For each group, ten ceca were processed per experiment for qRT-PCR. Cytokine mRNA expression levels were ascertained using RT-PCR TaqMan based assay with 28S as the reference gene. The RNAs were stored at −80 °C until plate setup. The cytokines IL-1β, IL-6, IL-8, IL-17, and IFN-γ were quantified utilizing a previously published method [16]. The primer and probe sequences for amplification have been previously published and described [17,18]. The plates were run in the Applied Biosystems ABI StepOne Plus PCR system (ThermoFisher Scientific, Waltham, MA, USA) with the previously stated TaqMan Assay under the following conditions: one cycle of 48 °C for 30 min, 95 °C for 20 s, and 40 cycles of 95 °C for 3 s and 60 °C for 30 s. Results were calculated with the corrected 40- C_t_ method [16] and expressed in fold change values. Each sample was run in triplicate for technical replication.

### 2.6. Data Analysis: Kinome Array

Once the peptide microarrays finished running (n = 27 total for the three tested time points), data normalization was performed for the kinome array using the PIIKA2 online platform (http://saphire.usask.ca/saphire/piika/index.html), a tool designed for in silico analysis of phosphorylation sites [19,20]. The array data were analyzed by conducting variance stabilization normalization and then performing a t-test, clustering, and pathway analysis for statistical data. Gene ontology (GO) and Kyoto Encyclopedia of Genes and Genomes (KEGG) pathway analyses were performed by uploading the statistically significant peptide lists to the Search Tool for the Retrieval of Interacting Genes (STRING) [21]. The protein-protein interaction networks were filtered by the minimum required interaction score with high confidence (0.700), in which the thickness of the connection lines indicate the strength of the data support.

## 3. Results

Broiler chicks fed 1% ChT had upregulated proinflammatory cytokine, specifically IL-6, mRNA expression in the ceca. Figure 1A,B depict each experimental fold change value across different days from the mRNA expression assay. The other tested pro-inflammatory cytokine results (IL-1β, IL-8, IL-17, IFN-γ) can be found in Appendix A. Of the proinflammatory cytokines tested, only IL-6 was found to be statistically significant in fold change differences (more than 32-fold) on days 6 and 10 in birds fed 1% ChT compared to the controls. On day 4, none of the cytokines tested was considered significant in fold change. By day 10, IL-6 still had a statistically significant three-fold change compared to the control, but not as elevated as it was on day 6.

For the kinome array, one of the major phosphorylated pathways identified to be affected by ChT was the JAK-STAT signaling pathway for all days tested. Appendix A display the complete interaction network of unique proteins from the kinome array for days 4, 6, and 10. The clustered peptides indicate closely related connections. These figures also reveal the number of links between immune pathways (those without links are not confidently linked as interactions according to STRING-db). For each day tested, the JAK-STAT pathway was listed as one of the most altered immune pathways, as highlighted in Table 1). To further evaluate the phosphorylation events, the peptides involved directly or indirectly in the JAK-STAT pathways are listed in Table 2. JAK1 was significantly altered (*p* ≤ 0.01) on day 6 with no change in phosphorylation state on days 4 and 10 (as indicated by the hyphen). JAK2 (*p* ≤ 0.01) also appeared to be a target for altered phosphorylation, as seen on days 4 and 6 in 1% ChT-fed chicks compared to the control. Another peptide of interest is the IL6ST peptide (Interleukin-6 receptor subunit beta), which had altered phosphorylation on days 4 and 10 (*p* < 0.05). The array also identified significant altered phosphorylation of STAT1, 3, and 5B activities on all tested days (*p* < 0.05). Therefore, the increased presence of JAK and STAT peptides indicated a stimulation in increased IL6 to stimulate JAK-STAT signaling pathway due to the inclusion of 1% ChT in the diet.

The other affected immune pathways identified on the kinome array were T-cell receptor signaling pathway and chemokine signaling pathway. The resulting phosphorylation events within these pathways are listed in Table 3 and Table 4. For the chemokine signaling pathway, CCR2 is one of the starting peptides of the pathway. This peptide was only significantly increased in phosphorylation on day 4, but no changes seen on days 6 and 10. Overall, neither the T-cell receptor nor the chemokine pathway showed significant activation by providing 1% ChT to broilers, as shown in Table 3 and Table 4.

## 4. Discussion

Demands for alternatives to antibiotics, such as ChT, in livestock feed have increased globally in the past several years. Growing research is showing the importance of the gastrointestinal system and its role in host immune modulation [22,23,24]. Numerous reviews have already been published regarding the evidence of ChT in improving overall host health and immunity during pathogenic events, but relatively little is known about the functional role that ChT play in immunity in a non-infection model. One of the most altered peptide pathways from the present study is the JAK-STAT signaling pathway, which is crucial for growth, energy, immunity, and overall development of the animal. This pathway is also important for its connection to IL-6 regulation, which was heavily upregulated on day 6 of the mRNA expression array (Figure 1A,B). Appendix A display the complete interaction network of unique proteins from the kinome array for days 4, 6, and 10. This network reveals the strong patterns towards immune connections, especially with the JAK-STAT signaling pathway (affiliated peptides indicated in the purple circles on each time point). Based on our findings, the kinomic analysis demonstrates the importance of a phytobiotic incorporation in feed, providing further evidence of its potential as a key ATA. These results may also provide useful information regarding how the JAK-STAT pathway is modified with the inclusion of phytobiotics in livestock diet.

The JAK family proteins are the initial receptors present to initiate the JAK-STAT pathway. In the chicken, JAK1 expression is found in the bursa and thymus at high levels, perhaps due to the post-translational modification activity of JAK1 kinase in these locations [25]. A previous study discovered interactions between avian JAK1 and STAT1/3/5B affected different cytokine families, including IL-6 and IL-10, in the gut during a disease state in the birds [26]. Another study in broiler chickens found upregulated JAK1 and STAT1,3,5B mRNA levels led to increased expressions of IL-2, IL-4, IL-6, IL-7, IL-10, IL-22, and IFN-α [26]. The mRNA expression data revealed an upregulation of statistically significant fold change on days 6 and 10, while the other proinflammatory cytokines were not found to be statistically significant. This provides evidence of an IL-6 immune mediated proinflammatory response, as opposed to initiating T-cell receptors or chemokine receptors. Although Table 1 shows that the T-cell receptor signaling pathway and chemokine signaling pathway are the top two pathways with most altered peptides, this information only provides altered peptide numbers and not detailed phosphorylation or dephosphorylation events. The chemokine signaling pathway appeared to follow the CCR2-GNB network of signaling pathways; however, with the significantly increased phosphorylation in STAT1,3,5B peptides and significant phosphorylation presence of JAK2, this points to even greater evidence of the JAK-STAT pathway being activated from the chemokine signaling pathway. As reflected by our mRNA expression results, chemokine IL-8 did not show any significant fold change between the 1% ChT group and the control group. The T-cell receptor pathway showed less clear pathway information, even though it was the immune pathway with the most altered peptides. However, this pathway is an important precursor pathway that promotes signaling cascades for other immune pathways, including the PI3K-Akt signaling pathway, the MAPK signaling pathway, and the NF-κβ signaling pathway [27]. As shown in Table 4, the direct peptides involved in the pathway do not indicate a clear T-cell function, which may be the reason why our IL-1 family cytokines did not show significant fold changes.

The kinome data provides further evidence of this trend with the decreasing phosphorylation events within the T-cell receptor signaling pathway and chemokine signaling pathway across all three days tested; see Table 3 and Table 4. The kinomic data for day 6 seem to follow the IL-6 family of cytokine signaling for the JAK-STAT pathway, shown in Appendix A: IL-6, which promotes the growth and differentiation of different cell types and can activate the pathway by phosphorylating JAK1,2 and then activating downstream signaling STAT family proteins, such as STAT1 and STAT3 [28,29]. Interestingly, the JAK-STAT signaling pathway was the only one that had IL6R out of the 13 signal pathways involved in avian embryonic stem cell differentiation [29]. A previous in vivo experiment suggested that IL-6 can regulate the JAK-STAT pathway by decreasing JAK2 and increasing levels of STAT3 [28,30], which is supported by our results. Our in vivo kinomic results, shown in Table 2, observed increasing phosphorylation of IL6ST on days 4 and 10, apart from dephosphorylation observed on day 6. This is especially noteworthy in the biological processes, in which phosphorylation of IL6ST at this site would alter signaling pathway regulation and transcription processes [31,32]. This dephosphorylation may be due to the host immune system attempting to regulate the strong pro-inflammatory response via the IL-6 mediated pathway. Notably, SOCS3, the primary inhibitor of IL-6 signaling, is present on days four and six in our data. The negative values as seen in this present study associated with the SOCS protein would indicate less presence of negative regulation of the JAK-STAT pathway [32,33,34]. This would potentially explain the negative values associated with the JAK2 and IL6ST on days four and six, respectively.

This study aimed to provide phosphorylation details regarding the broiler chick immune response when ChT are included into the grower diet. The ChT appear to be acting as an immune activator due to the high upregulation of IL-6 proinflammatory cytokine. In considering practical application, recent studies suggest the usage of encouraging proinflammatory states for beneficial purposes of stimulating the immune system at early stages to improve resistance against pathogens [23]. As a key signaling pathway involved in regulating innate immunity and adaptive immunity, the JAK-STAT pathway is crucial to study when considering using ATA such as ChT. Therefore, this study shows evidence of the modulatory effect of ChT on the host, suggesting a role of JAK-STAT pathway in altered immunomodulation via IL-6 signal transduction. Future experiments will investigate the usage of ChT as antimicrobial immunomodulators on the local cecal level.

## 5. Conclusions

Supplementing broiler diets on day 6 with 1% ChT provided evidence of modulation based on the alteration of JAK-STAT signaling pathway in the host. Based on the kinome array, the JAK-STAT pathway was one of the most altered signaling pathways involved in the immune response. This was further evidenced by the significant upregulation of IL-6 proinflammatory cytokine on day 6. This suggests a role of the JAK-STAT pathway in altered immunomodulation via IL-6 signal transduction pathway. However, more research will need to be conducted to further elucidate the effects of ChT on immunity and its role.

## Figures and Tables

**Figure 1 animals-11-00337-f001:**
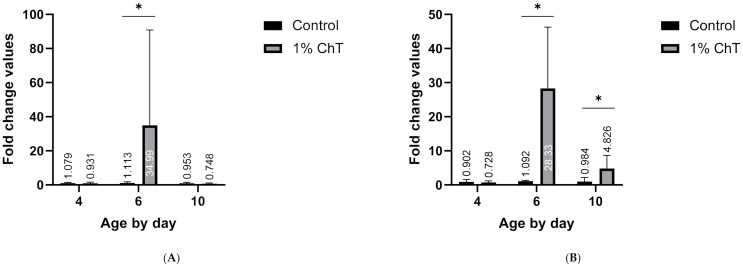
(**A**,**B**) IL-6 mRNA expression of cecal tissue from experimental chicks fed 1% ChT inclusion diet, determined by quantitative RT-PCR: (**A**) First experimental fold changes by age tested comparison of averaged 1% ChT inclusion group with control group; (**B**) Second experimental fold changes by age tested comparison of averaged 1% ChT inclusion group with control group. * = significantly different from the controls (*p* ≤ 0.05).

**Table 1 animals-11-00337-t001:** Summarized table of Kyoto Encyclopedia of Genes and Genomes (KEGG) immune pathways at days 4, 6, and 10 comparing 1% ChT-fed birds and control birds. The hyphens indicate fewer than ten peptides and a lack of significance for our analysis. The JAK-STAT signaling pathway has been highlighted.

Immune Pathways						
	Day 4		Day 6		Day 10	
Identified Pathways	Number of Peptides	*p*-Value	Number of Peptides	*p*-Value	Number of Peptides	*p*-Value
Chemokine signaling pathway	23	9.19 × 10^−18^	26	9.19 × 10^−19^	18	7.10 × 10^−13^
T cell receptor signaling pathway	14	7.55 × 10^−12^	25	5.85 × 10^−23^	12	7.50 × 10^−10^
Jak-STAT signaling pathway	13	1.38 × 10^−8^	19	7.82 × 10^−13^	15	1.31 × 10^−10^
B cell receptor signaling pathway	13	2.94 × 10^−12^	19	3.27 × 10^−18^	12	3.13 × 10^−11^
Fc gamma R-mediated phagocytosis	13	2.81 × 10^−11^	13	4.06 × 10^−10^	-	-
TNF signaling pathway	13	2.21 × 10^−10^	21	1.17 × 10^−17^	10	1.72 × 10^−7^
Wnt signaling pathway	13	4.24 × 10^−9^	11	3.30 × 10^−6^	-	-
Apoptosis	12	2.09 × 10^−8^	19	5.79 × 10^−14^	-	-
Toll-like receptor signaling pathway	11	1.46 × 10^−8^	20	6.03 × 10^−17^	13	9.43 × 10^−11^
Natural killer cell mediated cytotoxicity	11	8.57 × 10^−8^	16	1.76 × 10^−11^	-	-
NOD-like receptor signaling pathway	10	8.08 × 10^−6^	18	1.22 × 10^−11^	12	1.11 × 10^−7^
Th17 cell differentiation	10	1.43 × 10^−7^	22	3.36 × 10^−19^	10	1.11 × 10^−7^
IL-17 signaling pathway	-	-	16	3.62 × 10^−13^	-	-
Inflammatory mediator regulation of TRP channels	-	-	10	6.08 × 10^−7^	-	-

**Table 2 animals-11-00337-t002:** Proteins from the JAK-STAT signaling pathway that displayed a statistically significant change in phosphorylation. The UniProt ID helps identify the protein. Positive fold change values indicate phosphorylation events; negative values indicate dephosphorylation events. Only proteins that displayed a *p*-value of less than 0.05 are shown.

			JAK-STAT Signaling Pathway					
			Days of Necropsy					
			4		6		10	
Proteins	UniProt ID	p-Site	Fold Change	*p*-Value	Fold Change	*p*-Value	Fold Change	*p*-Value
AKT3	Q9Y243	S476/T305	−1.046	<0.01	−1.041	0.026	−1.023	0.034
CCND1	P24385	T286	-	-	1.071	0.015	-	-
EGFR	P00533	Y1069	-	-	1.032	0.041	-	-
EP300	Q09472	S89	-	-	-	-	1.052	0.031
GRB2	P62993	Y209	-	-	−1.063	0.02	1.044	0.017
IL12B	P29460	Y314	-	-	−1.048	0.03	-	-
IL6ST	P40189	Y915/Y759	1.058	0.026	−1.039	0.03	1.041	0.039
JAK1	P23458	Y1034	-	-	1.098	<0.01	-	-
JAK2	O60674	Y1007	−1.059	0.027	−1.042	<0.01	-	-
MTOR	P42345	S2481	1.061	0.042	−1.058	0.01	−1.041	0.036
PDGFRA	P16234	Y720	-	-	1.127	<0.01	−1.064	<0.01
PDGFRB	P09619	Y579	−1.115	<0.01	−1.061	0.019	−1.061	0.046
PIK3CB	P42338	Y425/S1070	1.086	<0.01	-	-	−1.049	0.022
PIK3R1	P27986	S608/Y467	1.058	0.028	−1.025	0.045	−1.076	<0.01
PIK3R2	O00459	Y365	1.052	0.038	−1.039	0.04	-	-
PIM1	P11309	S189	-	-	−1.046	<0.01	−1.043	0.036
RAF1	P04049	S259	1.053	0.03	−1.048	0.048	-	-
SOCS3	O14543	Y204	−1.069	<0.01	−1.052	<0.01	-	-
SOS1	Q07889	S1193	-	-	-	-	1.036	0.029
STAM2	O75886	Y371	-	-	-	-	−1.057	0.017
STAT1	P42224	Y701	1.057	0.048	1.060	0.033	1.034	0.034
STAT3	P40763	Y705/S727	1.067	0.015	1.078	0.031	1.046	0.044
STAT5B	P51692	Y699	−1.071	<0.01	1.061	<0.01	1.087	<0.01

**Table 3 animals-11-00337-t003:** Chemokine signaling pathway in chicken ceca on days 4, 6, and 10 of age. The results of this table indicate no clear path utilized by the host immunity of the chemokine signaling pathway other than presence of JAK2 and STAT proteins, which would suggest directionality toward the JAK-STAT pathway.

Chemokine Signaling Pathway			
Proteins	Day 4	Day 6	Day 10
AKT3	↓	↓	↓
ARRB1	↓	↓	-
ARRB2	-	↓	-
CHUK	-	↑	-
CRK	-	-	↑
CRKL	↓	-	-
CRR2	↑	-	-
GRB2	-	↓	↑
GRK5	↓	↓	-
GSK3A	↓	↑	↓
GSK3B	↓	↑	↑
JAK2	↓	↓	-
LYN	-	↑	-
MAP2K1	↑	-	-
NFKB1	-	↑	-
NFKBIA	-	↑	↓
PAK1	↓	-	-
PIK3R1	↑	↓	↓
PIK3R2	↑	↓	-
PRKCD	↓	↓	-
PTK2	↑	↑	↓
PTK2B	↑	↑	↑
PXN	↓	↑	↓
RAF1	↑	↓	-
SHC1	-	↑	-
SHC3	-	↑	-
SRC	-	↓	↑
STAT1	↑	↑	↑
STAT3	↑	↑	↑
STAT5B	↓	↑	↑

***↑*** = significantly (*p*
*≤* 0.05) increasing phosphorylated from noninfected control; *↓* = significantly (*p*
*≤* 0.05) decreasing phosphorylated from noninfected control; - = not significant from noninfected control.

**Table 4 animals-11-00337-t004:** T-cell receptor signaling pathway in chicken ceca on days 4, 6, and 10 of age. The results of this table indicate no clear path utilized by the host immunity of the T-cell receptor signaling pathway.

T Cell Receptor Signaling Pathway			
Proteins	Day 4	Day 6	Day 10
AKT3	↓	↓	↓
CHUK	-	↓	-
FOS	-	↑	-
FYN	↓	↓	-
GRB2	-	↓	↑
GSK3B	↓	↑	↑
JUN	↓	↓	-
LCK	-	↑	-
MAP2K1	↑	-	-
MAP2K2	-	↓	↑
MAP3K7	↓	↑	↑
NFATC1	↑	-	-
NFATC2	-	↓	-
NFATC3	↑	↓	↑
NFKB1	-	↑	-
NFKBIA	-	↑	↓
PAK1/2	↓	↓	-
PDPK1	↓	↓	-
PIK3R1	↑	↓	↓
PIK3R2	↑	↓	↓
PTPRC	-	↓	-
RAF1	↑	↓	-
ZAP70	-	↓	↓

*↑* = significantly (*p*
*≤* 0.05) increasing phosphorylated from noninfected control; *↓* = significantly (*p*
*≤* 0.05) decreasing phosphorylated from noninfected control; - = not significant from noninfected control.

## Data Availability

No new data were created or analyzed in this study. Data sharing is not applicable to this article.

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
