# Peer review of "Role of JAK-STAT Pathway in Broiler Chicks Fed with Chestnut Tannins"

_animals, 2021, doi:10.3390/ani11020337_

Round 1
Reviewer 1 Report
This study focus on the alternatives to antibiotic, this is really an important topic in the field of animal husbandry nowadays. The authors aimed to reveal the underlying mechanism of how chestnut tannins effect on chicks immunity. There is no problem with topic selection and experimental design, while the results need to further descript. The details were listed as follow:
- Line 31-32 and Table 2, there are formatting errors.
- Line59-60, supply the reference of the description “Our previous manuscript analyzed the metabolic phosphorylation events when ChT were included in the diet.”
- Line86, a total of 2, 4, 6, 8, and 10 days, 200 ceca were collected in this study, but why there is no results description of day 2 and 8 shown in the result section?
- Line108-110, descript the source of 10 cecal tissues used to extract RNA in detail, are these ceca stored in -80 ° C above mentioned, and then placed in the RNALater?
- This study identified several mRNA expression of inflammatory factor in cecal tissue from experimental chicks fed 1% ChT inclusion diet, but there was no proof or description that the cecum used in the experiment was healthy.
Author Response
- Line 31-32 and Table 2, there are formatting errors.
This was a formatting error when Animals transitioned my draft into this current form. Thank you for pointing this out.
- Line59-60, supply the reference of the description “Our previous manuscript analyzed the metabolic phosphorylation events when ChT were included in the diet.”
Thank you for this suggestion. It has been added to fix this.
- Line86, a total of 2, 4, 6, 8, and 10 days, 200 ceca were collected in this study, but why there is no results description of day 2 and 8 shown in the result section?
As mentioned in 2.3, we selected three days out of the 5 necropsy days due to our gene expression data from the previous manuscript. This was also an extremely costly analysis so we couldn’t run all necropsied tissues. I hope this makes sense!
- Line108-110, descript the source of 10 cecal tissues used to extract RNA in detail, are these ceca stored in -80 ° C above mentioned, and then placed in the RNALater?
No, these cecal tissues were directly placed into RNALater during necropsy. The ones stored in -80 C were used for the kinome array.
- This study identified several mRNA expression of inflammatory factor in cecal tissue from experimental chicks fed 1% ChT inclusion diet, but there was no proof or description that the cecum used in the experiment was healthy.
During the necropsies, we have not noted any abnormal ceca, including any abcesses or problematic cecal contents. In order to reduce any variables, we’ve received these birds from a reputable commercial hatchery with a certification of health (vaccines and health records provided) and ensured. Furthermore, we’ve allocated a small amount of cecal content to be spread onto differential agar for pathogen detection (full methodology in previously cited paper) for confirmation purposes.
Reviewer 2 Report
Dear Corresponding Author,
Please, find all reviewer comments, and queries below, and answer as details as possible.
Comment 1 - The title: the reviewer suggest changing "chicks" into "broiler chickens". It is more precise.
Comment 2 - Please, explain in detail the fact of the collaboration of the Authors with the Silvateam company. In the Conflict of Interest (L284-285), there is mentioned that no conflict occurred. Please, explain the role of Michele Battaglia. There is no information in the Author Contribution (L286-289). Due to this, it is not clear.
L12 - "Simple Summary" - it should be bolded?
L15 - ChT instead of chestnut tannins
L31-32 - there is a problem with editing, i.e., additional enter button in the middle of "significantly", please, correct.
L32 -if the Authors mentioned significant changes the exact p-value should appear; please, check the whole Abstract section in this case.
Comment 3 - (L25-27)"A total of 200 male day-of-hatch Cobb 500 chicks were randomly assigned to two treatment groups, totaling 50 chicks per pen per experiment. The treatments were as follows: (1) control feed – normal starter feed (n=50) and (2) 1% ChT inclusion feed (n=50)." - it is not clear; two groups, so 100 birds per group, with 50 birds per replication pen, thus 2 replications was used. The term "per experiment" is not correct... the current study includes only 1 experiment. It should be corrected.
Introduction
Comment 4 - The reviewer is not sure that citation editing is made properly. Please, double-check this in the Guidance to Authors.
Materiał and Methods
L62 - "at young age." the reviewer suggest changing this phrase to exact broiler age... in general, throughout the whole birds rearing animals are young... the time of rearing last only 35 d of age.
L64 - ChT instead of full name
L71-73 - "A total of 200 male day-of-hatch 71 Cobb 500 chicks were obtained from a local commercial hatchery and assigned to two 72 treatment groups, totaling 50 chicks per pen per experiment." - the same as above, please correct accordingly.
L74 - "normal starter feed" - there is no "normal" or "standard" diet in the broiler nutrition. Please, be preceise. Furthermore, the additional Table with the composition of the diet, as well as nutritive value is absolutely welcome, even as a supplementary material. Please, add information about the addition of other feed additives, such as coccidiostat, enzymes, etc.
L80 - ChT instead of chestnut tannin
L85 - "10 birds/group were euthanized", information about how the birds were chosen should be provided.
L86 - "each separate experiment" - still it is not clear... please, explain what the experiment means. It should be understandable to future readers.
L88-89 - "A total 88 of 200 ceca were collected at the end of both repeated studies." -because the replication within the study is not defined as an experimental unit, please, add information on what was defined as an exp. unit (bird?ceca?or the mean of both ceca?) and what was used in the statistical calculations.
Comment 5 - Data Analysis -there is a lack of information about the normal distribution of the data. How the Authors calculate the normal distribution. It is crucial to decide which test, i.e., parametrical or not, should be used further, i.e., t-test or Wilcoxon. Please, add this information. What kind of statistical software was used to compute the statistical analyses? Additionally, in this section it is crucial to add information about the replication number (n = ?) for each trait, as well as what was defined as an experimental factor.
Results
Comment 6- in the whole section the p-value should be added in the text where it is necessary (significant changes).
L133 - "As has been reported previously," please, delete.
L151 - pease, use shortcut ChT
Table 1 - the p-value should be unified with Table 2.
Table 2 - the table should be corrected in the case of editing.
Table 3 and 4 - there is another possibility to show the obtained data? maybe a graph?
Discussion
L202-203, L241 - the description of the all results should be located in the Results section not in the Discussion. Thus, whole supplementary material should be describe in the Results section. Please, correct.
Author Response
Comment 1 - The title: the reviewer suggest changing "chicks" into "broiler chickens". It is more precise.
Noted and changed!
Comment 2 - Please, explain in detail the fact of the collaboration of the Authors with the Silvateam company. In the Conflict of Interest (L284-285), there is mentioned that no conflict occurred. Please, explain the role of Michele Battaglia. There is no information in the Author Contribution (L286-289). Due to this, it is not clear.
He is a collaborator on the paper so I will go ahead and add his contribution. I did not catch that. Thank you!
L12 - "Simple Summary" - it should be bolded? Fixed.
L15 - ChT instead of chestnut tannins. Fixed
L31-32 - there is a problem with editing, i.e., additional enter button in the middle of "significantly", please, correct. Fixed. A lot of these are formatting issues once Animals switched my draft over. I thank you for catching these formatting errors.
L32 -if the Authors mentioned significant changes the exact p-value should appear; please, check the whole Abstract section in this case.
Since this is a generalized statement for the entire results, we feel we did not need to add the p-values since this will go over the maximum character limit. The main point of what we wanted to go over is highlighted in the abstract.
Comment 3 - (L25-27)"A total of 200 male day-of-hatch Cobb 500 chicks were randomly assigned to two treatment groups, totaling 50 chicks per pen per experiment. The treatments were as follows: (1) control feed – normal starter feed (n=50) and (2) 1% ChT inclusion feed (n=50)." - it is not clear; two groups, so 100 birds per group, with 50 birds per replication pen, thus 2 replications was used. The term "per experiment" is not correct... the current study includes only 1 experiment. It should be corrected.
Thank you for this note. I realized that I mentioned the two separate experimental replications performed in the main Materials and methods but not in the abstract. I have added to correct this error.
Introduction
Comment 4 - The reviewer is not sure that citation editing is made properly. Please, double-check this in the Guidance to Authors.
Will do.
Materiał and Methods
L62 - "at young age." the reviewer suggest changing this phrase to exact broiler age... in general, throughout the whole birds rearing animals are young... the time of rearing last only 35 d of age.
Fixed
L64 - ChT instead of full name
Fixed
L71-73 - "A total of 200 male day-of-hatch 71 Cobb 500 chicks were obtained from a local commercial hatchery and assigned to two 72 treatment groups, totaling 50 chicks per pen per experiment." - the same as above, please correct accordingly.
It is stated at the end of 2.1 that there were a total of two separate experiments performed
L74 - "normal starter feed" - there is no "normal" or "standard" diet in the broiler nutrition. Please, be preceise. Furthermore, the additional Table with the composition of the diet, as well as nutritive value is absolutely welcome, even as a supplementary material. Please, add information about the addition of other feed additives, such as coccidiostat, enzymes, etc.
I’ve changed it to “corn/soy” starter. I’ll have an added table of the nutritional information.
L80 - ChT instead of chestnut tannin
Fixed
L85 - "10 birds/group were euthanized", information about how the birds were chosen should be provided.
Fixed
L86 - "each separate experiment" - still it is not clear... please, explain what the experiment means. It should be understandable to future readers.
The experiment refers to an experimental replicate. I’ve had issues where the word “replicate” confuses readers and reviewers in the past, depending on their background. Therefore, the word experiment seems to resonate better overall.
L88-89 - "A total 88 of 200 ceca were collected at the end of both repeated studies." -because the replication within the study is not defined as an experimental unit, please, add information on what was defined as an exp. unit (bird?ceca?or the mean of both ceca?) and what was used in the statistical calculations.
I’ve expanded on this
Comment 5 - Data Analysis -there is a lack of information about the normal distribution of the data. How the Authors calculate the normal distribution. It is crucial to decide which test, i.e., parametrical or not, should be used further, i.e., t-test or Wilcoxon. Please, add this information. What kind of statistical software was used to compute the statistical analyses? Additionally, in this section it is crucial to add information about the replication number (n = ?) for each trait, as well as what was defined as an experimental factor.
As mentioned in the text, the normalization is performed on the PIIKA2 software. It’s an integrated software with built-in stats function for this type of analysis. I’ve mentioned the usage of t-test already. I will, however, add the replication number for clarification.
Results
Comment 6- in the whole section the p-value should be added in the text where it is necessary (significant changes).
They have been added where appropriate
L133 - "As has been reported previously," please, delete.
Fixed
L151 - pease, use shortcut ChT
Fixed
Table 1 - the p-value should be unified with Table 2.
I am a bit confused by your comment. What exactly do you mean by unified? I would appreciate any further clarification by the reviewer
Table 2 - the table should be corrected in the case of editing.
I apologize. The format shifted when Animals uploaded this draft for the reviewers.
Table 3 and 4 - there is another possibility to show the obtained data? maybe a graph?
The reason why we decided to format it this way is for the simplistic display of the data. Since these were not as significantly changed according to the kinome analysis, we focused more of the paper on the JAK-STAT pathway with the display of numerical data on a separate table.
Discussion
L202-203, L241 - the description of the all results should be located in the Results section not in the Discussion. Thus, whole supplementary material should be describe in the Results section. Please, correct.
The authors and I are recapping the results in this first paragraph which isn’t uncommon in discussions. We prefer this current format to summarize important points so it flows better for the reader. I hope that is alright with you. Please let me know if you'd like to discuss further.
Round 2
Reviewer 1 Report
Add the results or description of the pathogen detection in the “Materials and Methods” would be better.
Author Response
This section has been included